# Hospitalizations and length of stay of cancer patients: A cohort study in the Brazilian Public Health System

**Flávia Feliciana Silva**[1]*, **Gisele Macedo da Silva Bonfante**[2], **Ilka Afonso Reis**[3],
**Hugo André da Rocha**[1], **Agner Pereira Lana**[1], **Mariangela Leal Cherchiglia**[4]

**1** Medicine School, Postgraduate Program in Public Health, Federal University of Minas Gerais, Belo Horizonte, Brazil, **2** Department of Odontology, Pontifical Catholic University of Minas Gerais, Belo Horizonte, Brazil, **3** Department of Statistics, Institute of Exact Sciences, Federal University of Minas Gerais, Belo Horizonte, Brazil, **4** Department of Social and Preventive Medicine, Postgraduate Program in Public Health, Medicine School, Federal University of Minas Gerais, Belo Horizonte, Brazil

* flaviafelisil@yahoo.com.br

## Abstract

The hospitalizations are part of cancer care and has been studied by researchers world-wide. A better understanding about their associated factors may help to achieve improvements on this area. The aims of this study were to investigate the association between demographic and clinical characteristics and hospitalizations, as well as between these characteristics and the length of stay (LOS), within the first year of outpatient treatment, for the most incident cancers in the Brazilian population. In this cohort study, we investigated 417,477 patients aged 19 years or more, who started outpatient cancer treatment, from 2010–2014, for breast, prostate, colorectal, cervix, lung and stomach cancers. The outcomes evaluated were: i) Hospitalizations within the first year of outpatient cancer treatment; and ii) LOS of the hospitalized patients. It was performed a binary logistic regression to evaluate the association between the explanatory variables and the hospitalizations and a negative binomial regression to evaluate their influence on the length of hospital stay. The hospitalizations occurred for 34% of patients, with a median of LOS of 6 days (IQR: 2–15). Female patients were 16% less likely to be hospitalized (OR: 0.84; 95% CI: 0.82–0.86), with lower average of LOS (AR: 0.98; 95% CI: 0.97–0.99), each additional year of age reduced in 2% the hospitalization odds (OR: 0.98; 95% CI: 0.98–0.99) and in 1% the average of LOS (AR: 0.99; 95% CI: 0.98–0.99), patients from South region had twice more chances of hospitalization than from North region (OR: 2.01; 95% CI: 1.93–2.10) and patients with colorectal cancer had greater probability of hospitalization (OR: 4.42; 95% CI: 4.27–4.48), with the highest average of LOS (AR: 1.37; 95% CI: 1.35–1.40). In view of our results, we consider that the government must expand the policies with potential to reduce the number of hospitalizations.

**Data Availability Statement:** All relevant data are within the manuscript.

**Funding:** MLC received research sponsorship from National Council of Technological and Scientific Development (CNPq), Brazil (3060/2018-7) and the Research Support Foundation of the State of Minas Gerais (FAPEMIG), Brazil (PPM-00369-17). This study was part financed by the Coordination for the Improvement of Higher Education Personnel (CAPES) (Financial Code 001) and the Research Support Foundation of the State of Minas Gerais (FAPEMIG) grant number: APQ-03475-13 (MLC). The funders had no role in study design, data collection and analysis, decision to publish, or preparation of the manuscript.

## Introduction

Cancer is one of the main causes of worldwide morbidity and mortality [1]. In Brazil, the late diagnosis of this disease emphasizes a major public health problem [1–3]. The cancer diagnosis at advanced stages reduces the possibility of cure, increases patient's vulnerability to clinical complications and raises the need for higher technological support during treatment, which results a greater economic burden [4].

The Brazilian Public Health System (SUS) is configured as a decentralized network of health services that offers primary, secondary, and hospital care for free, across the Brazilian territory and to its entire population [2]. Almost 75% of the population resort only to the SUS when need to health care. The system reflects the Brazilian Constitutional Act that preconizes the health care as a universal right and as a State obligation. Despite this, the Brazil is affected by severe socioeconomic disparities, and its health system often suffers from chronic underfunding and reduced access in poorer regions [5]. Furthermore, the current healthcare model has been shown to be fragmented, centered on hospital care and mostly focused on the acute demands by the population [2,5].

In Brazil, the cancer treatments are performed in specialized care units, in both the public and private sectors. According the current system regulation, these services should be distributed among administrative health regions, according to population criteria [6]. The SUS finances the major part of cancer care in Brazil, due the large costs of treatment [7]. In addition, even some people who has private health insurance (about 25% of the Brazilian population) often perform their cancer treatment by the SUS [5].

The hospital admissions are part of the care trajectory of cancer patients [8], thus increasing the costs of treatment [9,10], especially in the last days of life [10]. In Brazil, the hospitalizations for cancer care cost about US$ 3 million annually, and presents a growth trend [11,12]. Along the treatment, some patients may suffer complications related to previous illnesses, acute clinical conditions or even worsening of cancer. These events influence the frequency of hospitalizations and, once admitted, the length of stay (LOS) [13].

The hospitalizations of cancer patients have been studied by researchers worldwide, from different perspectives. The results varied widely, depending on the study type and the population investigated. Some studies assessed the proportion of unplanned hospitalizations among cancer patients, which results ranged between 7% in Italy [8]; 45% to 58.8% in Australia [14,15]; and 59% to 67% in the United States [16,17]. Regarding the LOS, the findings varied slightly between these studies, being 5 days (mean) in Italy [8], 4.7 days (median) in Australia [15], and 7.4 days (mean) in the United States [16].

Regarding the frequency of hospitalizations among end-of-life cancer patients, a study conducted in seven developed countries (Belgium, Canada, England, Norway, Netherlands, United States and Germany) found that between 69.9% and 88.7% of the patients were hospitalized within 6 months before death [18]. When considering the readmissions of cancer patients, a study realized in Italy [19] and another in the United States [13] verified that 8.4% and 16.1% of patients, respectively, had 3 or more hospitalizations. The median of LOS observed was 9 days in the Italian study [19], and 4 days in the research of the USA [13]. Yet about this matter, it was verified that the LOS within Japanese hospitals was longer than in another developed countries, leading Japan to seek for ways to reduce the length of hospital stay [20,21].

It is important to emphasize that not always is observed some proportionality between frequency of hospitalizations and length of stay. A cancer patient may be hospitalized once, but to remain hospitalized for a long period, while others are hospitalized several times with short length of stay. The hospitalization and LOS are relevant indicators of health and quality of care for cancer patients [9,13,20].

In recent years, there has been an increase of studies among cancer patients that aimed to investigate the relationship between hospitalizations and the individual characteristics, also the attributes of the healthcare systems. The studies are heterogeneous, with poor possibility to generalize the results, since they evaluated populations with very particular characteristics, such as: elderly patients [18], individuals diagnosed with advanced cancer [13,22] or with very specific cancer types [16], patients treated on Intensive Care Unit or who underwent some specific treatment such as chemotherapy [15,23] or radiotherapy [24]. Many studies have focused on cost assessment, seeking to describe the potentially preventable hospitalizations [17]. In addition, many studies are restricted to populations in developed countries, such as the United States [18], Canada and Australia [14], and Europe [8,19].

Until now, we have not found any population-based study that investigate the factors associated with hospitalization among cancer patients in Brazil. A better understanding about the characteristics of patients undergoing cancer treatment and the factors associated with their hospital admissions may afford us to develop actions aiming to improve the health care, besides to contribute to the current scientific knowledge. Thus, the aims of this study were to investigate the association between demographic and clinical characteristics and hospitalizations, as well as between these characteristics and the length of stay (LOS), within the first year of outpatient treatment, for the most incident cancers in the Brazilian population.

## Methods

This research consists a non-concurrent cohort study. As data source, we used the National Oncological Database, a population-based cohort which contains all records of patients under oncological treatment in the SUS, from 2001 to 2015. This database is a subset of the National Database of Health, a data set centered on the individual that was built by record linkage techniques used to integrate data from the major SUS Information Systems: Outpatient (SIA), Hospital (SIH) and Mortality (SIM); from 2000 to 2015 [25].

We included 417,477 patients aged 19 years or more, who initiated outpatient cancer treatment at SUS, between October 1, 2010 and October 31, 2014, with the following primary tumor locations: breast, prostate, colon and rectum, cervix, lung and bronchi, and stomach; according to the International Classification of Disease, 10th Revision (ICD-10). These types were chosen since, apart from non-melanoma skin cancer, they are the most incident cancer types among the Brazilian population [1,26].

The clinical outcomes evaluated were: i) Hospitalizations within the first year after initiate the outpatient cancer treatment (chemotherapy or radiotherapy). Thus, considering the entire study population, the individuals were divided as 'hospitalized' or 'non-hospitalized'; ii) Length of Stay—LOS (in days), for those patients who were hospitalized. This variable was measured by the sum of days between the hospital admission and discharge of each hospitalization.

Given that our database ends in October 2015, we included patients who started treatment until October 2014. Thus, at least one year of follow-up was guaranteed for each patient in the study and there was no loss to follow-up, except from those who died before to complete one year of treatment.

The exposure variables analyzed were: i) Demographic characteristics: sex (male/female), age (years), age range (19–24, 25–39, 40–59, 60–79, ≥80 y/o), geographic region of residence (North, Northeast, Southeast, South and Midwest); ii) Clinical characteristics: primary tumor location (breast, prostate, colon and rectum, cervix, lung and bronchi, stomach), cancer stage at diagnosis (0, I, II, III, IV), number of comorbidities (0, 1–3, >3). The cancer stage was classified according to the TNM classification of malignant tumors, by the Union for International

Cancer Control [27], which ranges from 0 to IV. Regarding the 'length of stay' outcome analysis, the variable emergency hospitalization (Yes / No) was included.

The variable 'hospitalization cause' was used only in the descriptive analysis, since it correlates with the characteristics of the other clinical patients, and the regression model requires of the explanatory variables to be independent. The 'hospitalization cause' was observed from the code of procedure performed at the hospital, according to the SUS Procedures, Medicines and OPM Management System (SIGTAP) [11]. Thus, the causes of hospitalization were grouped as follows: oncological surgeries, clinical complications related to cancer, other clinical complications, chemotherapy/radiotherapy and other surgeries (surgical procedures non-specific for oncology). The hospitalizations caused by clinical complications related to cancer include acute complications, predictable or not, due to the malignant neoplasia or its treatment, and which requires hospitalization. Anemias, myelodysplastic syndromes, nausea and vomiting, pneumonia, hemorrhage, cachexia, sepsis and seizures are examples of complications included in this classification [11].

Yet about the descriptive analysis, the patient's characteristics were reported by frequency distributions for the categorical variables and the groups (hospitalized or non-hospitalized) were compared with the chi-square test. The age of patients (in years) was reported by median and interquartile range (IQR) and the comparison between the groups was realized by the Mann-Whitney test. The same approach was used to report the patient's follow-up time (in days) and it was also included the mean and standard deviation (SD).

Regarding the statistical analysis, it was performed a binary logistic regression model to evaluate the association between demographic and clinical variables and the occurrence of hospitalization. Thereby, we estimated the odds ratio (OR), considering a 95% confidence interval (95% CI). The goodness of fit was assessed by the Receiver Operating Characteristic (ROC) curve. In addition, considering the hospitalized patients, we realized a negative binomial regression model to evaluate the influence of the explanatory variables on the length of hospital stay (LOS). Thus, it was estimated the average ratio (AR), with a 95% CI. Both the statistical analyses were performed by univariate and multivariate models.

Considering that the variable 'LOS' is a count, there were two models to apply: the Poisson's and Negative Binomial models. Given that the dispersion of this variable was not adequate to the Poisson's distribution, we opted to the Negative Binomial model, which has a parameter that controls the over or under dispersion of data. The analyses were performed using the R Project for Statistical Computing software, version 3.4.4.

This study was part of the project "Epidemiological, economics and care paths of high cost procedures in SUS: use of patient-centered database from the integration of health information system records", approved by the Research and Ethics Committee of the Federal University de Minas Gerais, with Process number: CAAE 44121315.2.0000.5149.

## Results

The Table 1 describes the study population according to demographic and clinical characteristics. We included 417,477 patients who started outpatient cancer treatment by the SUS, between 2010 and 2014. The hospitalization within the first year of treatment occurred to 142,061 patients (34% of study population). Thus, 66% of patients (n = 275,416) were included in the non-hospitalized group. Considering the entire population of study, the majority was female (60.1%), with 62 y/o median age (IQR: 51–71 y/o) and residents of the Southeast region (47.5%). The most were treated for breast (37.8%) and prostate cancer (25.2%), diagnosed on advanced stages (III and IV) (53.3%) and had at least one comorbidity at treatment start (83%). Lastly, 13.7% of the study population died during the follow-up, which presented 338 days of mean time (SD: 77.8).

**Table 1. Demographic and clinical characteristics of cancer patients treated by the Brazilian Public Health System (SUS), 2010–2014.**

| Characteristic | Total patients | | Hospitalization | | | | p-value |
|---|---|---|---|---|---|---|---|
| | | | No | | Yes | | |
| | n = 417,477 | (100.%) | n = 275,416 | (66.0%) | n = 142,061 | (34.0%) | |
| **Gender** | | | | | | | <0.001 |
| Male | 166,626 | (39.9) | 113,895 | (41.4) | 52,731 | (37.1) | |
| Female | 250,851 | (60.1) | 161,521 | (58.6) | 89,330 | (62.9) | |
| **Age** (in years) * | | | | | | | |
| Median (IQR) | 62 | (51–71) | 59 | (49–69) | 63 | (52–72) | <0.001 |
| 19–24 | 1,031 | (0.2) | 438 | (0.2) | 593 | (0.4) | <0.001 |
| 25–39 | 28,652 | (6.9) | 16,240 | (5.9) | 12,412 | (8.7) | |
| 40–59 | 155,129 | (37.2) | 97,054 | (35.2) | 58,075 | (40.9) | |
| 60–79 | 201,336 | (48.2) | 138,890 | (50.4) | 62,446 | (44.0) | |
| ≥ 80 | 31,329 | (7.5) | 22,794 | (8.3) | 8,535 | (6.0) | |
| **Region of residence** | | | | | | | <0.001 |
| North | 17,121 | (4.1) | 12,773 | (4.6) | 4,348 | (3.1) | |
| Northeast | 91,535 | (21.9) | 63,896 | (23.2) | 27,639 | (19.5) | |
| Southeast | 198,249 | (47.5) | 131,623 | (47.8) | 66,626 | (46.9) | |
| South | 84,828 | (20.3) | 50,662 | (18.4) | 34,166 | (24.1) | |
| Midwest | 25,744 | (6.2) | 16,462 | (6.0) | 9,282 | (6.5) | |
| **Primary tumor location** | | | | | | | <0.001 |
| Prostate | 105,153 | (25.2) | 87,190 | (31.7) | 17,963 | (12.6) | |
| Breast | 157,959 | (37.8) | 110,204 | (40.0) | 47,755 | (33.6) | |
| Cervix | 42,837 | (10.3) | 27,934 | (10.1) | 14,903 | (10.5) | |
| Stomach | 22,532 | (5.4) | 10,241 | (3.7) | 12,291 | (8.7) | |
| Lung and Bronchi | 30,022 | (7.2) | 13,078 | (4.7) | 16,944 | (11.9) | |
| Colon and Rectum | 58,974 | (14.1) | 26,769 | (9.7) | 32,205 | (22.7) | |
| **Cancer stage** | | | | | | | <0.001 |
| 0 (in situ) | 19,449 | (4.7) | 13,809 | (5.0) | 5,640 | (4.0) | |
| I | 52,472 | (12.6) | 42,689 | (15.5) | 9,783 | (6.9) | |
| II | 123,155 | (29.5) | 93,723 | (34.0) | 29,432 | (20.7) | |
| III | 137,555 | (32.9) | 79,482 | (28.9) | 58,073 | (40.9) | |
| IV | 84,846 | (20.3) | 45,713 | (16.6) | 39,133 | (27.5) | |
| **Comorbidities (n)** | | | | | | | <0.001 |
| None | 70,784 | (17.0) | 64,113 | (23.3) | 6,671 | (4.7) | |
| 1–3 | 180,044 | (43.1) | 126,539 | (45.9) | 53,505 | (37.7) | |
| > 3 | 166,649 | (39.9) | 84,764 | (30.8) | 81,885 | (57.6) | |
| **Deaths**** | | | | | | | <0.001 |
| Yes | 57,281 | (13.7) | 16,445 | (5.9) | 40,836 | (28.7) | |
| No | 360,196 | (86.3) | 258,971 | (94.1) | 101,225 | (71.3) | |
| **Follow-up time** (in days) * | | | | | | | <0.001 |
| Mean (SD) | 338 | (77.8) | 352 | (56.7) | 310 | (101.9) | |
| Median (IQR) | 365 | (365–365) | 365 | (365–365) | 365 | (309–365) | |

IQR, interquartile range. SD, standard deviation. p-value estimated by The Chi-square and Mann-Whitney test (*).

**Deaths during the 1st year after outpatient treatment start.

Among the hospitalized patients, the most were female (62.9%), elderly (median age: 63 y/o; IQR: 52–72) and resident of Southeast region (46.9%). About the age range, we observed a lower proportion of elderly in the hospitalized group (50% with ≥60 y/o), when compared to

non-hospitalized patients (58.7% with ≥60 y/o). More than half of hospitalized patients underwent treatment for breast and colorectal cancer (56.3%), were diagnosed on advanced stages (III and IV) (68.4%) and had 4 or more comorbidities (57.6%). This group also presented almost 4-fold more deaths (28.7%) than the non-hospitalized (5.9%). Finally, the mean follow-up time of hospitalized patients was 310 days (SD: 101.9), against 352 days (SD: 56.7) of non-hospitalized, according to Table 1.

Regarding the hospitalizations, we observed a mean of 2.4 admissions per patient during the first year after outpatient treatment start, with 6 days of median of LOS (IQR: 2–15). When evaluating the characteristics of hospitalizations, presented in Table 2, we verified that most of the patients attended at least one emergency hospitalization during the study period (61.2%). The oncological surgeries have motivated majority of hospitalizations (30.3%), followed by the hospitalizations due to cancer-related clinical complications (25.7%) and other clinical causes (18.6%). Finally, the cancer type with more hospitalizations was the colon and rectum (33.1%), followed by breast cancer (24.8), which means that together these types registered more than half of the hospital admissions.

Considering the hospitalization as an outcome variable, the logistic regression analysis suggested that, among the cancer patients who started outpatient treatment between 2010 and 2014, the women were 16% less likely to be hospitalized (OR: 0.84; 95% CI: 0.82–0.86), the increase of one year in age reduced 2% the odds of hospitalization (OR: 0.98; 95% CI: 0.98–0.99) and patients from all residence regions had greater chances to be hospitalized when compared to the North region patients, as presented in Table 3.

Furthermore, the patients who live in the South region had twice hospitalization chances than North region patients (OR: 2.01; 95% CI: 1.93–2.10). Regarding the primary tumor location, patients with cancer of 'colon and rectum' (OR: 4.42; 95% CI: 4.27–4.48), 'stomach' (OR: 4.41; 95% CI: 4.29–4.53) and 'lung and bronchi' (OR: 4.35; 95% CI: 4.21–4.48) had greater probability of hospitalization. Also, the hospitalization odds increased as higher was the cancer stage at diagnosis, being the patients with stage IV more likely to be hospitalized in 62% (OR: 1.62; 95% CI:

**Table 2. Characteristics of hospitalizations of cancer patients treated by the Brazilian Public Health System (SUS), 2010–2014.**

| Characteristic | n = 342,734 | (100%) |
|---|---|---|
| **Hospitalization type** | | |
| Emergency | 209,702 | (61.2) |
| Elective | 133,032 | (38.8) |
| **Hospitalization cause** | | |
| Oncological surgeries | 103,965 | (30.3) |
| Cancer-related clinical complications | 88,088 | (25.7) |
| Other clinical causes | 63,781 | (18.6) |
| Chemotherapy/radiotherapy | 56,409 | (16.5) |
| Other surgeries | 30,491 | (8.9) |
| **Patient's cancer type** | | |
| Colon and Rectum | 113,546 | (33.1) |
| Breast | 85,083 | (24.8) |
| Stomach | 39,268 | (11.5) |
| Lung and Bronchi | 37,313 | (10.9) |
| Cervix | 35,081 | (10.2) |
| Prostate | 32,443 | (9.5) |

IQR, interquartile range. SD, standard deviation.

**Table 3. Logistic regression analysis of cancer patient's hospitalizations in the first year of treatment by Brazilian Public Health System (SUS), 2010–2014.**

| Characteristic | Simple model | | | Multiple model | | |
|---|---|---|---|---|---|---|
| | OR | 95% CI | p-value | OR | 95% CI | p-value |
| **Gender** | | | <0.001 | | | <0.001 |
| Male | - | - | - | - | - | - |
| Female | 1.19 | 1.17–1.21 | | 0.84 | 0.82–0.86 | |
| **Age** (in years) | 0.98 | 0.98–0.98 | <0.001 | 0.98 | 0.98–0.99 | <0.001 |
| **Region of residence** | | | <0.001 | | | <0.001 |
| North | - | - | - | - | - | - |
| Northeast | 1.27 | 1.22–1.31 | | 1.38 | 1.33–1.44 | |
| Southeast | 1.48 | 1.43–1.54 | | 1.43 | 1.38–1.49 | |
| South | 1.98 | 1.91–2.05 | | 2.01 | 1.93–2.10 | |
| Midwest | 1.65 | 1.58–1.72 | | 1.75 | 1.67–1.83 | |
| **Primary tumor location** | | | <0.001 | | | <0.001 |
| Prostate | - | - | - | - | - | - |
| Breast | 2.10 | 2.06–2.14 | | 2.42 | 2.34–2.50 | |
| Cervix | 2.59 | 2.52–2.65 | | 2.61 | 2.51–2.71 | |
| Stomach | 5.84 | 5.70–5.97 | | 4.41 | 4.29–4.53 | |
| Lung and Bronchi | 6.28 | 6.11–6.46 | | 4.35 | 4.21–4.48 | |
| Colon and Rectum | 5.82 | 5.65–6.00 | | 4.42 | 4.27–4.48 | |
| **Cancer stage** | | | <0.001 | | | <0.001 |
| 0 (in situ) | - | - | - | - | - | - |
| I | 0.56 | 0.54–0.58 | | 0.56 | 0.54–0.58 | |
| II | 0.76 | 0.74–0.79 | | 0.78 | 0.75–0.81 | |
| III | 1.78 | 1.73–1.84 | | 1.55 | 1.50–1.61 | |
| IV | 2.09 | 2.02–2.16 | | 1.62 | 1.57–1.70 | |
| **Comorbidities (n)** | 1.20 | 1.20–1.21 | <0.001 | 1.19 | 1.18–1.19 | <0.001 |

OR, odds ratio; 95% CI, 95% confidence interval.

1.57–1.70). Lastly, each additional comorbidity has risen by 19% the odds of hospitalization in the first year after treatment start (OR: 1.19; 95% CI: 1.18–1.19), according to Table 3.

The area under the ROC curve for the logistic regression model, after adjustment of the study variables (AUC measure), presented a value of 0.76, which demonstrates the good performance of the model.

The Table 4 presents the estimates of the negative binomial regression, which was performed to evaluate the influence of explanatory variables over the length of stay (LOS). The demographic factors associated with the highest averages of LOS were: to be male (AR: 0.98 for women; 95% CI: 0.97–0.99); with younger ages, since each additional year has reduced by 1% the average LOS (AR: 0.99; 95% CI: 0.98–0.99); and to live in the North region. About the clinical characteristics, the greatest averages of LOS were registered to patients with 'primary tumor location' in the colon and rectum (AR: 1.37; 95% CI: 1.35–1.40) and cervix (AR: 1.35; 95% CI: 1.32–1.38); with advanced cancer stage (stage IV) (AR: 1.04; 95% CI: 1.04–1.06); and that have been emergency hospitalized (AR: 1.28; 95% CI: 1.27–1.28). Lastly, each additional comorbidity increased by 3% the average of LOS (AR: 1.03; 95% CI: 1.02–1.03).

## Discussion

This study evaluated the factors associated with hospitalization and the length of hospital stay of cancer patients, within the first year after outpatient treatment start. We included 417,477

**Table 4. Negative binomial regression analysis of cancer patient's Length of Stay (LOS) in the first year of treatment by Brazilian Public Health System (SUS), 2010–2014.**

| Characteristic | Simple model | | | Multiple model | | |
|---|---|---|---|---|---|---|
| | AR | 95% CI | p-value | AR | 95% CI | p-value |
| **Gender** | | | <0.001 | | | 0.032 |
| Male | - | - | - | - | - | |
| Female | 0.76 | 0.75–0.77 | | 0.98 | 0.97–0.99 | |
| **Age** (*in years*) | 0.99 | 0.98–0.99 | <0.001 | 0.99 | 0.98–0.99 | <0.001 |
| **Region of residence** | | | <0.001 | | | <0.001 |
| North | - | - | - | - | - | |
| Northeast | 0.94 | 0.90–0.97 | | 0.93 | 0.93–0.98 | |
| Southeast | 0.92 | 0.89–0.95 | | 0.81 | 0.79–0.83 | |
| South | 0.99 | 0.96–1.03 | 0.866 | 0.75 | 0.73–0.77 | |
| Midwest | 0.80 | 0.77–0.83 | | 0.66 | 0.64–0.68 | |
| **Primary tumor location** | | | <0.001 | | | <0.001 |
| Prostate | - | - | - | - | - | |
| Breast | 0.58 | 0.57–0.59 | | 0.68 | 0.66–0.69 | |
| Cervix | 1.50 | 1.47–1.53 | | 1.35 | 1.32–1.38 | |
| Stomach | 1.48 | 1.45–1.52 | | 1.27 | 1.25–1.30 | |
| Lung and Bronchi | 1.36 | 1.33–1.38 | | 1.22 | 1.20–1.25 | |
| Colon and Rectum | 1.53 | 1.50–1.56 | | 1.37 | 1.35–1.40 | |
| **Cancer stage** | | | <0.001 | | | <0.001 |
| 0 (in situ) | - | - | - | - | - | |
| I | 0.84 | 0.81–0.87 | | 0.92 | 0.89–0.95 | |
| II | 0.89 | 0.86–0.91 | | 0.91 | 0.89–0.94 | |
| III | 0.94 | 0.92–0.97 | | 0.95 | 0.93–0.97 | |
| IV | 1.24 | 1.21–1.28 | | 1.04 | 1.01–1.06 | |
| **Emergency hospitalization (n)** | 1.32 | 1.32–1.33 | <0.001 | 1.28 | 1.27–1.28 | <0.001 |
| **Comorbidities (n)** | 1.06 | 1.05–1.06 | <0.001 | 1.03 | 1.02–1.03 | <0.001 |

AR, average ratio; 95% CI, 95% confidence interval.

patients that underwent cancer treatment by the Brazilian Public Health System (SUS), between 2010 and 2014, considering the most incident cancer types in Brazil, namely: breast, prostate, colon and rectum, cervix, lung and bronchi, and stomach. In the overview of our study population, we observed a predominance of female patients, elderly, residents of the Southeast region, treated for breast cancer, diagnosed on advanced stages and with comorbidities at treatment start. The hospitalization within the first year of treatment occurred for 34% of patients, from which the median of length of stay was 3 days (IQR: 2.0–7.0). In addition, we observed that several characteristics were associated with the hospitalization and length of stay, such as: gender, patient's age, geographic region of residence, primary tumor location, cancer stages at diagnosis, number of comorbidities at treatment start and emergency hospitalization.

In relation to the predominance of women in our study population, in recent years, the Brazilian government has expanded the public policies for screening and diagnosis of cervical and breast cancer, thus increasing the female population attendance in health services to treat these cancer types [12]. Concerning about the patient's age observed in our results, the epidemiological characteristics of the cancer types on this study explain the large proportion of elderly patients (> 60 y/o) [11,28]. Regarding the population distribution, our findings about the geographic regions agrees with the stats of IBGE (Brazilian Institute of Geography and Statistics),

given that the Southeast and Northeast regions contain the largest part of the Brazilian population. However, considering the hospitalized patients, the South region presented a higher proportion of individuals when compared to the Northeast region, which may be related to a greatest supply of hospital beds in the South region [28]. In relation to the primary tumor location, the Brazilian demographic changes and consequent population aging are associated to the increasing prevalence of breast and prostate cancer [1,26]. The large proportion of patients with advanced stages of cancer can be explained by the frequently late diagnosis and delay on treatment start, which represents a major public health issue in Brazil [1,26]. About the comorbidities, it is known that about 80% of Brazilians with 60 years or more have at least one morbidity [3].

About the hospitalizations and the length of stay of cancer patients, we observed that 34% of the individuals were hospitalized within the first year of outpatient treatment, and the median of LOS was 6 days (IQR: 2–15). Some related researches presented varied results, depending on the study type and the population investigated. Regarding the proportion of unplanned hospitalizations among cancer patients, the results ranged between 7% in Italy [8]; 45% to 58.8% in Australia [14,15]; and 59% to 67% in the United States [16,17]. In relation to the length of stay (LOS), the same studies showed slightly different findings, being 5 days (mean) in Italy [8], 4.7 days (median) in Australia [15], and 7.4 days (mean) in the United States [16]. In addition, when evaluating the hospital readmissions of cancer patients, it was verified that in Italy 8.4% of patients had 3 or more hospitalizations [19], while in the USA this proportion was 16.1% [13]. The median of LOS observed was 9 days in Italy [19] and 4 days in the USA [13]. Lastly, other studies noticed that the LOS within Japanese hospitals was longer when compared with some developed countries, leading Japan to seek for ways to reduce the length of hospital stay [20,21].

The often search by cancer patients for emergency services due to acute conditions, which resulted in hospital admissions, was also discussed by some authors [8,13,14,21]. A suggested explain argues that in the cancer treatment there would be a weak relationship between the patients and their healthcare team, or their contact with the reference professionals would be limited [4]. The most cited pathologies that took cancer patients to search for emergency care correspond to the procedure codes referred to the patient's clinical complications evaluated in this study, which corroborates to our findings [8,29]. Given that some of these clinical complications are preventable, its suggested that an integrated healthcare, with professionals from diverse areas, could improve the quality of care and avoid unnecessary hospitalizations [30].

Regarding the evaluation of factors associated to the hospitalization and length of stay, some studies presented a predominance of male patients, with worse clinical severity and less survival among men [21,31,32]. In this study, the adjustment of the variables for statistical analysis would require us to assume other factors, and more researches would be needed to better understand the hospitalization patterns among male patients.

About the influence of patient's age in our study, the reduced odds of hospitalization and lower average LOS observed among elderly patients differ from other studies that analyzed this association [21,31–33]. As the patient's age increases, some pre-existing clinical conditions get worse, the toxicity of drugs aggravates and the biological response to illnesses becomes slower [8,30]. Therefore, our results for the patient's age variable is controversial, given that the elderly patients frequently demand emergency units in the SUS [34].

The current model of emergency care in Brazil, with the adoption of Emergency Care Units (UPAs) as an intermediate service between primary and hospital care, may support to explain the reduced odds of hospitalization and the inferior LOS among elderly people. The patients admitted in the UPAs must remain in this service for a maximum of 24 hours, while waiting for hospital beds available by the SUS. However, the lack of hospital beds forces the patients to long stays in these units [34,35]. In Brazil, this situation is so critical that some health managers

consider the beds of UPAs as hospital beds and, sometimes, as intensive care units, even without the necessary resources for this type of care. In addition, many UPAs are not computerized, thus the length of stay in these services is not accounted by the Hospital Information System (SIH), since the payment for the healthcare provided by UPAs is not realized by an Authorization for hospital admittance (AIH) [35].

Regarding the geographic region of patient's residence, it is known that the North region has the lowest supply of hospital beds in Brazil. Moreover, there has been observed a gradual decrease of the number of hospital beds in the North region, which reduced from 2.25 (in 1992) to 1.84 beds per 100,000 inhabitants (in 2010) [28]. Therefore, this evidence may justify the reduced odds of hospitalization among the cancer patients from this region. Yet about this matter, the North region presents the worst indicators of cancer care in Brazil. None of the levels of care hold enough resources, including the hospital institutions. The commonly late diagnosis and consequent delay to treatment start, even among cancers targeted by public policies, may support an explanation to the highest LOS observed among the North region patients [1,12,30].

The regional inequalities in the Brazilian health system have also been evidenced by the analysis of palliative care implementation, which is poorly consolidated in the North region when compared, as example, with the Southeast. The most developed cities, such as São Paulo, has been pioneers in this strategy, leading to the early discharge of patients, since it allows the cancer treatment by support of home care teams [36]. These qualified teams are capable to monitor cancer patients at home, in interface with the hospital staff, thus configuring one way to reduce the length of hospital stay [21,36]. The insufficiency of this kind of care and of structure resources within the public health system, which are needed to support patients with terminal illnesses, may give us a comprehension about the search for hospital services at the end-of-life, resulting in longer stays and higher expenses with hospitalizations [4,10].

The higher odds of hospital admissions registered among patients with primary tumors in colon and rectum, stomach, and lung are justified by the clinical aspects of these cancers, above all for those associated with complications due to surgical treatment [17,21,24,32,33]. Furthermore, as observed in our results, other studies verified longer length of hospital stay (LOS) among patients with colorectal cancer [14,32,33].

Regarding the cancer stages, the worse clinical conditions of patients with metastatic tumors may require higher LOS for pain relief, due to hemodynamic instability or even for emotional support to patients and their families, since the cancer cure is not possible, which is an argument that corroborates our study results [17,21,33].

In relation to patient's comorbidities, as observed in our results, it was expected that the increase of its number would rise the odds of hospitalization and extend the average length of stay. The stabilization of acute conditions due to comorbidities, by itself, may requires the hospitalization [9]. Once the patient is hospitalized, the presence of comorbidities increases the risk of complications, then influences the planned time for the hospital discharge [9,20,31].

The emergency hospitalization of cancer patients presupposes that these individuals present some serious pathologies that does not allow their outpatient control [8,13,21,29]. In most of the cases, emergency hospitalizations are assisted by general practitioners, unlinked to the patient's referral health team or even without expertise on oncology [30]. Under these conditions, the evaluation to determine an accurate diagnosis and properly treatment may demand longer hospital stays for these patients [32].

## Limitations

The main limitations of this study are related to the administrative character of our database, since it presents some gaps of clinical information and individual characteristics, such as

socioeconomic variables. Regarding this, the patients who underwent only surgery as treatment were not included in the study, due their lack of information of cancer stage at treatment start. The absence of data from these patients must be mentioned, given that the surgery is commonly the recommended treatment modality for patients on early stages of cancer. However, these limitations are overcome by the benefits of using a large database that includes the entire population of patients treated for cancer in the Brazilian Public Health System (SUS).

## Conclusions

Throughout this study, we observed several factors associated with the hospitalizations, as well as with the length of stay (LOS), of cancer patients treated by the Brazilian Public Health System (SUS). Among the variables associated with higher attendance of hospitalizations and with longer LOS, we shall highlight the 'primary tumor location' (lung and bronchi, stomach, colon and rectum) and 'cancer stage' at treatment start (III and IV). About the patient's residence region, we observed great discrepancies between the North and South regions. Although controversial, we observed that older patients had reduced chances of hospitalization, with shorter LOS. Also, most of the patients underwent emergency hospitalizations, which may suggest the worsening of disease or side effects of drugs used on treatment.

In view of our results, we consider that the government must expand the policies with potential to reduce the number of hospitalizations, such as the provision of palliative care. This strategy is fragmented in Brazil, which holds a low number of service providers. Finally, the present study contributes to evaluate the scenario of cancer care in Brazil, given the insufficient public hospital care, caused in overall by the historical lack of funding to the SUS. Only by investments in public policies, the Brazilian government may improve the health access with equity for its every citizen.

## Author Contributions

**Conceptualization:** Flávia Feliciana Silva, Gisele Macedo da Silva Bonfante, Mariangela Leal Cherchiglia.

**Data curation:** Flávia Feliciana Silva, Hugo André da Rocha, Agner Pereira Lana, Mariangela Leal Cherchiglia.

**Formal analysis:** Flávia Feliciana Silva, Ilka Afonso Reis, Hugo André da Rocha, Agner Pereira Lana.

**Methodology:** Flávia Feliciana Silva, Gisele Macedo da Silva Bonfante, Ilka Afonso Reis, Hugo André da Rocha, Mariangela Leal Cherchiglia.

**Supervision:** Mariangela Leal Cherchiglia.

**Writing – original draft:** Flávia Feliciana Silva.

**Writing – review & editing:** Flávia Feliciana Silva, Gisele Macedo da Silva Bonfante, Ilka Afonso Reis, Hugo André da Rocha, Agner Pereira Lana, Mariangela Leal Cherchiglia.

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
