## [Decision Letter · Decision Letter 0]

28 Nov 2019

PONE-D-19-27629

PATIENTS  HOSPITALIZATIONS AND LENGTH OF HOSPITAL STAY  WITH   CANCER TREATMENT BY BRAZILIAN PUBLIC HEALTH SYSTEM

PLOS ONE

Dear Dr Silva,

Thank you for submitting your manuscript to PLOS ONE. After careful consideration, we feel that it has merit but does not fully meet PLOS ONE’s publication criteria as it currently stands. Therefore, we invite you to submit a revised version of the manuscript that addresses the points raised during the review process.

ACADEMIC EDITOR:   

Sorry for my delay in decision making. It was very difficult for me to find reviewers. 

In my personal opinion the manuscript is not presented in a standard English, so I would appreciate a revision by an English native speaker. 

There are some conflicts between reviewers. I think the statistical analysis should be improved. Please try to answer to all the criticisms expecially revising the statistical method. PLOS requires a rigorous and thorough statistical analysis. 

We would appreciate receiving your revised manuscript by dec 19th. To enhance the reproducibility of your results, we recommend that if applicable you deposit your laboratory protocols in protocols.io, where a protocol can be assigned its own identifier (DOI) such that it can be cited independently in the future. For instructions see: http://journals.plos.org/plosone/s/submission-guidelines#loc-laboratory-protocols

We look forward to receiving your revised manuscript.

Kind regards,

Martina Crivellari

Academic Editor

PLOS ONE

1. In ethics statement in the manuscript and in the online submission form, please provide additional information about the patient records used in your retrospective study. Specifically, please ensure that you have discussed whether all data were fully anonymized before you accessed them and/or whether the IRB or ethics committee waived the requirement for informed consent. If patients provided informed written consent to have data from their medical records used in research, please include this information.

2. Thank you for including the following funding information within the acknowledgments section of your manuscript; "MLC, EIGA and FAA received research sponsorship from National Council of Technological and Scientific Development (CNPq), Brazil and The Minas Gerais State Research Foundation (FAPEMIG), Brazil. This study was part financed by the Coordination for the Improvement of Higher Education Personnel (CAPES) (Financial Code 001) and Fundação de Amparo à Pesquisa do Estado de Minas Gerais (FAPEMIG) grant number: APQ-03475-13."

Reviewers' comments:

Reviewer's Responses to Questions

**Comments to the Author**

1. Is the manuscript technically sound, and do the data support the conclusions?

Reviewer #1: Partly

Reviewer #2: Partly

2. Has the statistical analysis been performed appropriately and rigorously? 

Reviewer #1: No

Reviewer #2: Yes

3. Have the authors made all data underlying the findings in their manuscript fully available?

Reviewer #1: Yes

Reviewer #2: Yes

4. Is the manuscript presented in an intelligible fashion and written in standard English?

Reviewer #1: No

Reviewer #2: Yes

5. Review Comments to the Author

Reviewer #1: In this retrospective observational study, Silva and colleagues analyse the percentage of hospitalization and the length of hospital stay among Brazilian cancer patients within the first year of treatment, during a 4-year period, from 2010 to 2014.

The main strength of this study is the large sample size, reporting the data of more than 400,000 patients, unfortunately the characteristics of Brazilian heath system and social conditions makes hard to generalize these findings, reducing the interest of this study for a worldwide audience.

Major comments:

1) English level is so poor, that makes difficult to understand the manuscript, please have a full language revision

2) The methods does not explain how missing data were addressed

3) Statistical methods are poorly described and sometimes incorrect:

- “Frequency measures were performed to describe the demographic…” which one? Please specify all variables’ type and description. It is not clear neither completed. For example, You do not describe continuous variables

- Age distribution is not normal, therefore it is better described by median and interquartile range (IR)

- it is not specified the test used to evaluate p value in Table 1

- length of stay should be described with median and IR

- Please describe better your primary outcome, and the reason because you perform more than one regression model.

- Please specify the reason because you use the RR estimation in the negative binomial regression.

- In addition, no criteria are detailed in order to put the variables in logistic and negative models.

4) the length of hospital stay in each category should be reported, you might add a column to Table 1

5) The decision to exclude cancer patients eligible to surgery as isolated treatment, excludes a large percentage of early-stage patients (especially for cervix, breast and prostate tumors). You should address this limitation properly in the discussion.

Minor comments:

1) The title does not specify what kind of study is this and it is grammatically incorrect

2) You also need to explain better in which way Brazilian Health System (what does it mean “bed regulation system”?)

3) The National Oncological Database in part of the National Database of Health? If this is the case, please clarify in the manuscript. Otherwise specify your data source correctly.

4) Among exclusion criteria what does it mean: “authorization and payments within the SUS”?

5) How can “hospitalization within the first year of treatment” be both a clinical outcome and a demographic characteristics?

Reviewer #2: All comments are in the attached file..........................................................................................................................................................................................................................................

6. PLOS authors have the option to publish the peer review history of their article (what does this mean?). If published, this will include your full peer review and any attached files.

Reviewer #1: No

Reviewer #2: Yes: Satar Rezaei, Ph.D

---

## [Author Response · Author response to Decision Letter 0]

16 Apr 2020

To editor

Comment: In my personal opinion the manuscript is not presented in a standard English, so I would appreciate a revision by an English native speaker.

Response: The article has undergone process of revision for the English language, as recommended.

Comment: I think the statistical analysis should be improved. Please try to answer to all the criticisms especially revising the statistical method. PLOS requires a rigorous and thorough statistical analysis.

Response: In order to improve the clarity and transparency of this study, we have revised and rewritten part of the Methods session, including Statistical methods.

Comment: In ethics statement in the manuscript and in the online submission form, please provide additional information about the patient records used in your retrospective study. Specifically, please ensure that you have discussed whether all data were fully anonymized before you accessed them and/or whether the IRB or ethics committee waived the requirement for informed consent. If patients provided informed written consent to have data from their medical records used in research, please include this information. 

Response: About this question, we put a Funding Statement section of the online submission form.

“This research was conducted in accordance to the regulatory guidelines and requirements of the National Health Council / Ministry of Health of Brazil n° 466/2012. The Institutional Review Board (IRB) at the Federal University of Minas Gerais -UFMG, approved the study (Process 44121315.2.0000.5149)”.

“In the linkage data, there were established strict access rules to ensure data security and the confidentiality of information. During the probabilistic linkage only, individual identification fields were included, ie, no additional sensitive information (eg. comorbidities or causes of death) was available. At the end of the linkage process, all identifying information has been deleted and a new unidentified ID field was created. Therefore, the database which had been used in the analyses is anonymous and de-identified”.

Comment: We note that you have provided funding information that is not currently declared in your Funding Statement. However, funding information should not appear in the Acknowledgments section or other areas of your manuscript. We will only publish funding information present in the Funding Statement section of the online submission form.

Response: We removed the funding-related text from the manuscript and updated our online Funding Statement. 

“Funding/Support: MLC received research sponsorship from National Council of Technological and Scientific Development (CNPq), Brazil and the Research Support Foundation of the State of Minas Gerais (FAPEMIG), Brazil. This study was part financed by the Coordination for the Improvement of Higher Education Personnel (CAPES) (Financial Code 001) and the Research Support Foundation of the State of Minas Gerais (FAPEMIG) grant number: APQ-03475-13. The funders had no role in study design, data collection and analysis, decision to publish, or preparation of the manuscript”

Reviewer(s)' Comments to Author: 

Thank you for accept to review our paper. We are sure that your contributions have allowed us to increase the study quality and we hope to have taken the best from all your questions and suggestions.

Reviewer: 1 

Comment: In this retrospective observational study, Silva and colleagues analyse the percentage of hospitalization and the length of hospital stay among Brazilian cancer patients within the first year of treatment, during a 4-year period, from 2010 to 2014. The main strength of this study is the large sample size, reporting the data of more than 400,000 patients, unfortunately the characteristics of Brazilian health system and social conditions makes hard to generalize these findings, reducing the interest of this study for a worldwide audience. 

Response: The Brazilian Public Health System (SUS) is configured as a decentralized network of health services that offers primary, secondary, and hospital care for free, across the Brazilian territory and to its entire population [1]. Almost 75% of the population resort only to the SUS when need to health care. The system reflects the Brazilian Constitutional Act that preconize the health care as a universal right and as a State obligation. It was loosely patterned after the National Health Services of European countries like the United Kingdom. Despite this, the Brazil is affected by severe socioeconomic disparities, and its health system often suffers from chronic underfunding and reduced access in poorer regions [2]. The SUS finances the major part of cancer care in Brazil, due the large costs of treatment [4]. In addition, even some people who has private health insurance (about 25% of the Brazilian population) often perform their cancer treatment by the SUS [3].

We conducted an unprecedented study in Brazil, including data from all patients treated by the SUS, between the years 2010 and 2014, for the types of cancer with the highest incidence in the country. In addition to the large sample size, as pointed out by the reviewer, one of the relevant points of this study was to cover patients from different regions of Brazil, contextualizing the hospitalization of cancer patients in locations with different social and economic conditions. The organization of a national health system presents particularities that cover socio-economic and cultural aspects of each nation and the sharing of information from diverse contexts contributes to the expansion of scientific knowledge. The results of the present study are believed to be robust and relevant, contributing to a better understanding of hospitalizations of cancer patients around the world. 

In addition, we added in the Introduction two paragraphs describing the SUS and its importance in Brazil (page 3, lines 63-77).

Major comments:

Comment: English level is so poor, that makes difficult to understand the manuscript, please have a full language revision

Response: The article has undergone process of revision for the English language, as recommended.

Comment: The methods does not explain how missing data were addressed.

Response: Considering this question, we put a sentence that explains why our study does not show missing, since all patients were followed up for one year (365 days), except for those who died during that period.

Methods (page 6; lines 149-151): “Thus, at least one year of follow-up was guaranteed for each patient in the study and there was no loss to follow-up, except from those who died before to complete one year of treatment.”

Comment: Statistical methods are poorly described and sometimes incorrect.

Response: In order to improve the clarity and transparency of this study, we have revised and rewritten part of the Methods session, including Statistical methods.

Comment: “Frequency measures were performed to describe the demographic…” which one? Please specify all variables’ type and description. It is not clear neither completed. For example, You do not describe continuous variables.

Response: As suggested we rewrote the paragraph that describe the descriptive analysis.

Methods (pages 7-8; lines 174-80): “Yet about the descriptive analysis, the patient's characteristics were reported by frequency distributions for the categorical variables and the groups (hospitalized or non-hospitalized) were compared with the chi-square test. The age of patients (in years) was reported by median and interquartile range (IQR) and the comparison between the groups was realized by the Mann-Whitney test. The same approach was used to report the patient's follow-up time (in days) and it was also included the mean and standard deviation (SD).”

Comment: Age distribution is not normal, therefore it is better described by median and interquartile range (IR)

Response: Suggestion accepted. In Table 1 (page 9, line 221) we put the variable age as median (interquartile range) and we describe it in the Results.

Results (page 9, lines 206-208; 212-213): “Considering the entire population of study, the majority was female (60,1%), with 62 y/o median age (IQR: 51-71 y/o) and residents of the Southeast region (47.5%).”…

…“Among the hospitalized patients, the most were female (62.9%), elderly (median age: 63 y/o; IQR: 52-72) and resident of Southeast region (46.9%)”.

Comment: it is not specified the test used to evaluate p value in Table 1.

Response: About this suggestion, we better described the descriptive analysis as mentioned above. We also put it as footnote in Table 1. (page 9, line 221).

Methods (pages 7-8; lines 174-180): “Yet about the descriptive analysis, the patient's characteristics were reported by frequency distributions for the categorical variables and the groups (hospitalized or non-hospitalized) were compared with the chi-square test. The age of patients (in years) was reported by median and interquartile range (IQR) and the comparison between the groups was realized by the Mann-Whitney test. The same approach was used to report the patient's follow-up time (in days) and it was also included the mean and standard deviation (SD).”

Comment: length of stay should be described with median and IR.

Response: Considering this suggestion, we reported the length of stay outcome by median and interquartile range (IQR).

Results (page 10, lines 222-223). “Regarding the hospitalizations, we observed a mean of 2.4 admissions per patient during the first year after outpatient treatment start, with 6 days of median of LOS (IQR: 2 - 15)”.

Comment: Please describe better your primary outcome, and the reason because you perform more than one regression model.

Response: About this suggestion, it seems that the two objectives of our study were not sufficiently clear, as well as the exposure variables analyzed. Therefore, we rewrote the objectives of the study and described better our two outcome variables in methods. Thus, we performed a binary logistic regression model for the first objective and a negative binomial regression model for the second objective.

Introduction (page 5, lines 123-127): “Thus, the aims of this study were to investigate the association between demographic and clinical characteristics and hospitalizations, as well as between these characteristics and the length of stay (LOS), within the first year of outpatient treatment, for the most incident cancers in the Brazilian population.”

Methods (page 6, lines 142-147): “The clinical outcomes evaluated were: i) Hospitalizations within the first year after initiate the outpatient cancer treatment (chemotherapy or radiotherapy). Thus, considering the entire study population, the individuals were divided as 'hospitalized' or 'non-hospitalized'; ii) Length of Stay - LOS (in days), for those patients who were hospitalized. This variable was measured by the sum of days between the hospital admission and discharge of each hospitalization.”

Methods (page 8, lines 181-193): “Regarding the statistical analysis, it was performed a binary logistic regression model to evaluate the association between demographic and clinical variables and the occurrence of hospitalization. Thereby, we estimated the odds ratio (OR), considering a 95% confidence interval (95% CI). The goodness of fit was assessed by the Receiver Operating Characteristic (ROC) curve. In addition, considering the hospitalized patients, we realized a negative binomial regression model to evaluate the influence of the explanatory variables on the length of hospital stay (LOS). Thus, it was estimated the average ratio (AR), with a 95% CI. Both the statistical analyses were performed by univariate and multivariate models”.

“Considering that the variable 'LOS’ is a count, there were two models to apply: the Poisson's and Negative Binomial models. Given that the dispersion of this variable was not adequate to the Poisson's distribution, we opted to the Negative Binomial model, which has a parameter that controls the over or under dispersion of data”.

Minor comments:

Comment: The title does not specify what kind of study is this and it is grammatically incorrect.

Response: Considering this suggestion, we were rewrite the title. 

Title (page 1, lines 4-5): “Hospitalizations and length of stay of patients with cancer: a cohort study in the Brazilian Public Health System.”

Comment: You also need to explain better in which way Brazilian Health System (what does it mean “bed regulation system”?)

Response: We introduced two paragraphs that better describe the Brazilian Public Health System.

Introduction (page 3, lines 63-77): “The Brazilian Public Health System (SUS) is configured as a decentralized network of health services that offers primary, secondary, and hospital care for free, across the Brazilian territory and to its entire population [2]. Almost 75% of the population resort only to the SUS when need to health care. The system reflects the Brazilian Constitutional Act that preconizes the health care as a universal right and as a State obligation. Despite this, the Brazil is affected by severe socioeconomic disparities, and its health system often suffers from chronic underfunding and reduced access in poorer regions [5]. Furthermore, the current healthcare model has been shown to be fragmented, centered on hospital care and mostly focused on the acute demands by the population [2,5].”

“In Brazil, the cancer treatments are performed in specialized care units, in both the public and private sectors. According the current system regulation, these services should be distributed among administrative health regions, according to population criteria [6]. The SUS finances the major part of cancer care in Brazil, due the large costs of treatment [7]. In addition, even some people who has private health insurance (about 25% of the Brazilian population) often perform their cancer treatment by the SUS [5].”

Comment: The National Oncological Database in part of the National Database of Health? If this is the case, please clarify in the manuscript. Otherwise specify your data source correctly.

Response: As suggested we better explained the paragraph that describes The National Oncological Database. 

Methods (page 6, lines 130-135): “As data source, we used the National Oncological Database, a population-based cohort which contains all records of patients under oncological treatment in the SUS, from 2001 to 2015. This database is a subset of the National Database of Health, a data set centered on the individual that was built by record linkage techniques used to integrate data from the major SUS Information Systems: Outpatient (SIA), Hospital (SIH) and Mortality (SIM); from 2000 to 2015 [25].”

Comment: Among exclusion criteria what does it mean: “authorization and payments within the SUS”?.

Response: In regard this suggestion, we rewrote the inclusion and exclusion criteria.

Methods (page 6, lines 136-141): “We included 417,477 patients aged 19 years or more, who initiated outpatient cancer treatment at SUS, between October 1, 2010 and October 31, 2014, with the following primary tumor locations: breast, prostate, colon and rectum, cervix, lung and bronchi, and stomach; according to the International Classification of Disease, 10th Revision (ICD-10). These types were chosen since, apart from non-melanoma skin cancer, they are the most incident cancer types among the Brazilian population [1,26].”

Comment: How can “hospitalization within the first year of treatment” be both a clinical outcome and a demographic characteristics?

Response: About this suggestion, as we explained above, it seems that the two objectives of our study were not sufficiently clear, as well as the exposure variables analyzed. Therefore, we rewrote the objectives of the study and the outcomes analyzed. Thus, we performed a binary logistic regression model for the first objective and a negative binomial regression model for the second objective. So, “hospitalization in the first year of treatment” is an exposure variable in the first analysis and a clinical characteristic in the second (). In addition, a new table was included in the paper (current Table 2), where we present the characteristics of hospitalizations of cancer patients treated by the Brazilian Public Health System (SUS), 2010-2014.

Introduction (page 5, lines 123-127): “Thus, the aims of this study were to investigate the association between demographic and clinical characteristics and hospitalizations, as well as between these characteristics and the length of stay (LOS), within the first year of outpatient treatment, for the most incident cancers in the Brazilian population.”

Methods (page 6, lines 142-147): “The clinical outcomes evaluated were: i) Hospitalizations within the first year after initiate the outpatient cancer treatment (chemotherapy or radiotherapy). Thus, considering the entire study population, the individuals were divided as 'hospitalized' or 'non-hospitalized'; ii) Length of Stay - LOS (in days), for those patients who were hospitalized. This variable was measured by the sum of days between the hospital admission and discharge of each hospitalization.”

Reviewer: 2 

Thank you for accept to review our paper. We are sure that your contributions have allowed us to increase the study quality and we hope to have taken the best from all your questions and suggestions.

Abstract

Comment: Please use the LOS as a standard abbreviation for this concept in all of text.

Response: Suggestion accept.

Abstract (page 2, lines 36-39): “The aims of this study were to investigate the association between demographic and clinical characteristics and hospitalizations, as well as between these characteristics and the length of stay (LOS), within the first year of outpatient treatment, for the most incident cancers in the Brazilian population.”

Comment: Why the SUS is used to public health system. it is not related. Please use PHS instead of SUS for the first time.

Response: In the Introduction, we replaced by Brazilian Public Health System (SUS). SUS is the commonly acronymous used to designate our health system in Portuguese.

Introduction (page 3, lines 63-65): “The Brazilian Public Health System (SUS) is configured as a decentralized network of health services that offers primary, secondary, and hospital care for free, across the Brazilian territory and to its entire population [2].”

Comment: Please replaced by 95% CI. 

Response: We removed this sentence and rewrote the abstract, adapting it to the number of words requested. 

Comment: RR<= is it unusual. Please check and correct it. 

Response: Suggestion accepted and correction done.

Comment: the conclusion part must be written based on the study findings. The present format is not associated with the study results. Please modify it..

Response: On this suggestion, we rewrote the Conclusions session, now properly based on the results of the study.

Abstract (page 2, lines 54-55) “In view of our results, we consider that the government must expand the policies with potential to reduce the number of hospitalizations”.

Conclusions (page 19-20, lines 397-413).

Introduction

Comment: The authors should be provide more detail information about health service delivery to international reader in the Introduction section.

Response: Considering this, we have written a paragraph describing how the SUS provides health service delivery for cancer treatment.

Introduction (page 3, lines 72-77): “In Brazil, the cancer treatments are performed in specialized care units, in both the public and private sectors. According the current system regulation, these services should be distributed among administrative health regions, according to population criteria [6]. The SUS finances the major part of cancer care in Brazil, due the large costs of treatment [7]. In addition, even some people who has private health insurance (about 25% of the Brazilian population) often perform their cancer treatment by the SUS [5].”

Comment: also, the main reasons for doing the studies topic is not provided. Please add some information about this issue at the last paragraph of Introduction section.

Response: About this suggestion, we wrote two paragraphs where we provide the main reasons for doing the studies topic.

Introduction (page 5; lines 108-127): “In recent years, there has been an increase of studies among cancer patients that aimed to investigate the relationship between hospitalizations and the individual characteristics, also the attributes of the healthcare systems. The studies are heterogeneous, with poor possibility to generalize the results, since they evaluated populations with very particular characteristics, such as: elderly patients [18], individuals diagnosed with advanced cancer [13,22] or with very specific cancer types [16], patients treated on Intensive Care Unit or who underwent some specific treatment such as chemotherapy [15,23] or radiotherapy [24]. Many studies have focused on cost assessment, seeking to describe the potentially preventable hospitalizations [17]. In addition, many studies are restricted to populations in developed countries, such as the United States [18], Canada and Australia [14], and Europe [8,19]”.

“Until now, we have not found any population-based study that investigate the factors associated with hospitalization among cancer patients in Brazil. A better understanding about the characteristics of patients undergoing cancer treatment and the factors associated with their hospital admissions may afford us to develop actions aiming to improve the health care, besides to contribute to the current scientific knowledge. Thus, the aims of this study were to investigate the association between demographic and clinical characteristics and hospitalizations, as well as between these characteristics and the length of stay (LOS), within the first year of outpatient treatment, for the most incident cancers in the Brazilian population”.

Comment: finally, it is suggested that author provide previous studies about the interest topic in other countries. there is no information about LOS and hospitalization rate among patients with cancer in the introduction section.

Response: Considering this suggestion, we included two paragraphs in the Introduction session, about previous studies on the interest topic in other countries, then we add these studies to the Discussion session.

Introduction (page 4; lines 85-102): “The hospitalizations of cancer patients have been studied by researchers worldwide, from different perspectives. The results varied widely, depending on the study type and the population investigated. Some studies assessed the proportion of unplanned hospitalizations among cancer patients, which results ranged between 7% in Italy [8]; 45% to 58.8% in Australia [14-15]; and 59% to 67% in the United States [16,17]. Regarding the LOS, the findings varied slightly between these studies, being 5 days (mean) in Italy [8], 4.7 days (median) in Australia [15], and 7.4 days (mean) in the United States [16]”.

“Regarding the frequency of hospitalizations among end-of-life cancer patients, a study conducted in seven developed countries (Belgium, Canada, England, Norway, Netherlands, United States and Germany) found that between 69.9% and 88.7% of the patients were hospitalized within 6 months before death [18]. When considering the readmissions of cancer patients, a study realized in Italy [19] and another in the United States [13] verified that 8.4% and 16.1% of patients, respectively, had 3 or more hospitalizations. The median of LOS observed was 9 days in the Italian study [19], and 4 days in the research of the USA [13]. Yet about this matter, it was verified that the LOS within Japanese hospitals was longer than in another developed countries, leading Japan to seek for ways to reduce the length of hospital stay [20,21]”.

Discussion (page 15, lines 296-310): “About the hospitalizations and the length of stay of cancer patients, we observed that 34% of the individuals were hospitalized within the first year of outpatient treatment, and the median of LOS was 6 days (IQR: 2 - 15). Some related researches presented varied results, depending on the study type and the population investigated. Regarding the proportion of unplanned hospitalizations among cancer patients, the results ranged between 7% in Italy [8]; 45% to 58.8% in Australia [14,15]; and 59% to 67% in the United States [16,17]. In relation to the length of stay (LOS), the same studies showed slightly different findings, being 5 days (mean) in Italy [8], 4.7 days (median) in Australia [15], and 7.4 days (mean) in the United States [16]. In addition, when evaluating the hospital readmissions of cancer patients, it was verified that in Italy 8.4% of patients had 3 or more hospitalizations [19], while in the USA this proportion was 16.1% [13]. The median of LOS observed was 9 days in Italy [19] and 4 days in the USA [13]. Lastly, other studies noticed that the LOS within Japanese hospitals was longer when compared with some developed countries, leading Japan to seek for ways to reduce the length of hospital stay [20,21]”.

Method

Comment: Why the authors did not use other social demographic variables in the analysis? Please clarify and provide more details about it. 

Response: We apologize for not presenting this information before. We inserted a paragraph with the limitations of the study, explaining why we did not use other sociodemographic variables in the analysis. 

Limitations (page 19, lines 387-395): “The main limitations of this study are related to the administrative character of our database, since it presents some gaps of clinical information and individual characteristics, such as socioeconomic variables. Regarding this, the patients who underwent only surgery as treatment were not included in the study, due their lack of information of cancer stage at treatment start. The absence of data from these patients must be mentioned, given that the surgery is commonly the recommended treatment modality for patients on early stages of cancer. However, these limitations are overcome by the benefits of using a large database that includes the entire population of patients treated for cancer in the Brazilian Public Health System (SUS).”

Comment: Please provide the criteria for the choosing of the NB model in this study as compared with Poisson model

Response: As suggested, we have rewritten the paragraph in the Methods session that explain why we use a negative binomial regression model instead of the Poisson’s regression model. 

Methods (page 8, lines 190-193): “Considering that the variable 'LOS’ is a count, there were two models to apply: the Poisson's and Negative Binomial models. Given that the dispersion of this variable was not adequate to the Poisson's distribution, we opted to the Negative Binomial model, which has a parameter that controls the over or under dispersion of data.”

Results

Comment: Table 1_ The subtitle and variables in this table in not full capital. please change it.

Response: Change done as suggested. 

Discussion

Comment: The limitation of the study is missing. Please add some limitation at the end of the conclusion part.

Response: We apologize for not presenting this information before. We inserted a paragraph with the limitations of the study.

Limitations (page 19, lines 387-395): “The main limitations of this study are related to the administrative character of our database, since it presents some gaps of clinical information and individual characteristics, such as socioeconomic variables. Regarding this, the patients who underwent only surgery as treatment were not included in the study, due their lack of information of cancer stage at treatment start. The absence of data from these patients must be mentioned, given that the surgery is commonly the recommended treatment modality for patients on early stages of cancer. However, these limitations are overcome by the benefits of using a large database that includes the entire population of patients treated for cancer in the Brazilian Public Health System (SUS).”

Conclusion

Comment: the convulsion part is not related to the results of the study. Please check and clarify it.

Response: On this suggestion, we rewrote the concluding paragraph, based on the results of the study.

Conclusion (page 19, lines 397-413): “Throughout this study, we observed several factors associated with the hospitalizations, as well as with the length of stay (LOS), of cancer patients treated by the Brazilian Public Health System (SUS). Among the variables associated with higher attendance of hospitalizations and with longer LOS, we shall highlight the 'primary tumor location' (lung and bronchi, stomach, colon and rectum) and 'cancer stage' at treatment start (III and IV). About the patient's residence region, we observed great discrepancies between the North and South regions. Although controversial, we observed that older patients had reduced chances of hospitalization, with shorter LOS. Also, most of the patients underwent emergency hospitalizations, which may suggest the worsening of disease or side effects of drugs used on treatment.”

“In view of our results, we consider that the government must expand the policies with potential to reduce the number of hospitalizations, such as the provision of palliative care. This strategy is fragmented in Brazil, which holds a low number of service providers. Finally, the present study contributes to evaluate the scenario of cancer care in Brazil, given the insufficient public hospital care, caused in overall by the historical lack of funding to the SUS. Only by investments in public policies, the Brazilian government may improve the health access with equity for its every citizen”.

---

## [Decision Letter · Decision Letter 1]

4 May 2020

Hospitalizations and length of stay of cancer patients: a cohort study in the Brazilian Public Health System

PONE-D-19-27629R1

Dear Dr. Silva,

We are pleased to inform you that your manuscript has been judged scientifically suitable for publication and will be formally accepted for publication once it complies with all outstanding technical requirements.

With kind regards,

Martina Crivellari

Academic Editor

PLOS ONE

Additional Editor Comments (optional):

Reviewers' comments:

Reviewer's Responses to Questions

**Comments to the Author**

1. If the authors have adequately addressed your comments raised in a previous round of review and you feel that this manuscript is now acceptable for publication, you may indicate that here to bypass the “Comments to the Author” section, enter your conflict of interest statement in the “Confidential to Editor” section, and submit your "Accept" recommendation.

Reviewer #2: All comments have been addressed

2. Is the manuscript technically sound, and do the data support the conclusions?

Reviewer #2: Yes

3. Has the statistical analysis been performed appropriately and rigorously? 

Reviewer #2: Yes

4. Have the authors made all data underlying the findings in their manuscript fully available?

Reviewer #2: Yes

5. Is the manuscript presented in an intelligible fashion and written in standard English?

Reviewer #2: Yes

6. Review Comments to the Author

Reviewer #2: Thank you to authors to address my my comments. ................................................................

7. PLOS authors have the option to publish the peer review history of their article (what does this mean?). If published, this will include your full peer review and any attached files.

Reviewer #2: No

---

## [Editor Report · Acceptance letter]

8 May 2020

PONE-D-19-27629R1 

Hospitalizations and length of stay of cancer patients: a cohort study in the Brazilian Public Health System 

Dear Dr. Feliciana Silva:

I am pleased to inform you that your manuscript has been deemed suitable for publication in PLOS ONE. Congratulations! Your manuscript is now with our production department. 

With kind regards,

on behalf of

Dr. Martina Crivellari 

Academic Editor

PLOS ONE